# Anterior Scleral Thickness and Anterior Segment Biometrics Measured with Swept Source Ocular Coherence Tomography in High Myopic Eyes with and without Glaucoma: A Comparative Study

**DOI:** 10.3390/diagnostics14060655

**Published:** 2024-03-20

**Authors:** Bachar Kudsieh, Rocio Vega-González, Sofia Bryan, Elena Almazan-Alonso, Mariluz Puertas, Lucia Gutiérrez-Martin, Ignacio Flores-Moreno, Jorge Ruiz-Medrano, Muhsen Samaan, Jose Maria Ruiz-Moreno

**Affiliations:** 1Department of Ophthalmology, University Hospital Puerta De Hierro Majadahonda, 28220 Madrid, Spain; 2Centro Internacional de Oftalmologia Avanzada, 28010 Madrid, Spain; 3Instituto de Microcirugia Ocular (IMO), 28035 Madrid, Spain; 4Barraquer Eye Clinic UAE, Dubai P.O. Box 212619, United Arab Emirates

**Keywords:** high myopia, glaucoma, anterior scleral thickness, Schlemm’s canal, trabecular meshwork, conjunctival thickness, anterior segment optical coherence tomography

## Abstract

**Background**: To assess the anterior scleral thickness (AST), Schlemm’s canal diameter (SCD), trabecular meshwork diameter (TMD) and conjunctiva tenon capsule thickness (CTT) in high myopic (HM) subjects and HM subjects with glaucoma (HMG) compared to control eyes. **Methods:** One hundred and twenty eyes were included, and AST at 0, 1, 2 and 3 mm from the scleral spur, SCD, TMD and CTT were measured. **Results**: Mean age was 64.2 ± 11.0 years, and the temporal SCD and temporal TMD were significantly longer in the HMG subjects compared to the controls (380.0 ± 62 μm vs. 316.7 ± 72 μm, *p* = 0.001) and (637.6 ± 113 μm vs. 512.1 ± 97 μm, *p* = 0.000), respectively. There were no significant differences between the HM and HMG subjects in SCD and TMD (all *p* > 0.025). Compared to the HM subjects, the temporal AST0 (432.5 ± 79 μm vs. 532.8 ± 99 μm, *p* = 0.000), temporal AST1 (383.9 ± 64 μm vs. 460.5 ± 80 μm, *p* = 0.000), temporal AST2 (404.0 ± 68 μm vs. 464.0 ± 88 μm, *p* = 0.006) and temporal AST3 (403.0 ± 80 μm vs. 458.1 ± 91 μm, *p* = 0.014) were significantly thinner in the HMG group. No differences were found between the CTT in the three groups (all *p* > 0.025). **Conclusions**: Our data indicate a thinner AST in HMG subjects and no differences in SCD and TMD between HM and HMG subjects.

## 1. Introduction

Myopia is the leading cause of visual impairment worldwide; its incidence is supposed to increase in the next few decades [1]. Glaucoma, myopic maculopathy and retinal detachment are the main causes of vision loss in highly myopic eyes [1]. Several studies showed that high myopia (HM) is a potential risk factor for primary open-angle glaucoma (POAG) [2,3]; additionally, individuals with HM usually have more pronounced glaucomatous optic neuropathy and greater visual field progression [4,5].

The sclera constitutes approximately 90% of the outer layer of the eye, providing the main support to retina, choroids and optic nerve head [6]; furthermore, it plays an important role in determining the biomechanical properties of the cornea [7,8]. Trabecular meshwork (TM) and Schlemm’s canal (SC), which forms the main part of aqueous humour outflow drainage system [9], are located between the cornea and anterior sclera; additionally, the scleral spur as a part of the anterior sclera has been reported to be a supporting tissue for TM and SC and a longer scleral spur could better maintain the morphology and the function of TM and SC [10,11]. In addition, the conjunctiva is a key ocular structure for eye anterior segment being a supportive tissue for aqueous humour outflow drainage system; recent studies have pointed out the conjunctival thickness as possible predicting factor of bleb functionality after glaucoma surgery [12,13].

Different imaging techniques have been used to assess, in vivo, the anterior segment biometrics of the eye, mostly via ultrasound biomicroscopy (UBM) and spectral domain optical coherence tomography (OCT) [14]. Nowadays, the OCT technology has evolved to the newer Swept Source (SS)-OCT technology that offers high resolution images and accurate measurements of the anterior segment structures, including aqueous humour outflow drainage system [15,16].

Various hypotheses have been proposed to explain the association between HM and glaucoma, but the exact mechanism is still unknown [17]. It has been demonstrated that HM eyes with larger SC dimensions and thinner TM were associated with reduced anterior scleral thickness (AST), as compared with emmetropic eyes [18]. However, to our knowledge, no previous studies have analysed the anterior segment biometrics of the eye in HM subjects with glaucoma (HMG) as compared to either healthy HM or non-myopic control eyes using SS-OCT. Moreover, most of studies were conducted on non-Caucasic subjects. Oliveira et al. [19] showed that AST was thinner in Caucasians than in non-Caucasians.

Hence, the current study was designed to assess the biometrics of the anterior segment (AS) in HM subjects with and without glaucoma as compared to control eyes, especially the AST, which plays a key role in the aqueous humour outflow and may be involved in the pathogenesis of glaucoma in HM subjects.

## 2. Materials and Methods

### 2.1. Setting and Participants

This cross-sectional study was conducted with 61 subjects that attended the ophthalmology department at Hospital Universitario Puerta de Hierro between 30 May 2022 and 30 May 2023. Subjects who met all the inclusion criteria and none of the exclusion criteria were invited to participate after providing written informed consent. The study protocol adhered to the tenets of the Declaration of Helsinki and was approved by the Hospital Review Board.

In the HM group, the inclusion criteria were Caucasian subjects with an axial length (AXL) > 26 mm. In this group, exclusion criteria were any concomitant ocular pathology, intraocular pressure (IOP) < 21 mmHg, previous ocular surgery excluding corneal refractive surgery and ocular surface diseases, such as pterygium or pinguecula, that may difficult the AS-OCT examination.

In the HMG group, the inclusion criteria were Caucasian subjects with an AXL > 26 mm, open angle visualized by gonioscopy, basal IOP without treatment ≥ 21 mmHg, glaucomatous optic discs and visual defects evaluated by two glaucoma experts (BK and LGM). Exclusion criteria were any concomitant ocular pathology, previous ocular surgery excluding corneal refractive surgery and ocular surface diseases, such as pterygium or pinguecula, that may difficult the AS-OCT examination.

Controls were defined as Caucasian subjects aged > 18 years with an AXL between 20.5 to 24 mm, IOP < 21 mmHg, open anterior chamber angle and normal fundus with no visible peripapillary retinal nerve fibre layer (RNFL) defects.

### 2.2. Ophthalmological Examination and OCT

All the included subjects underwent, on the same day, a medical history, a complete ophthalmologic examination and SS AS-OCT. The ophthalmologic exam included visual acuity, slit lamp biomicroscopy, AXL and anterior chamber depth (ACD) measured by IOL Master 500 (Carl Zeiss Meditec AG, Jena, Germany), tonometry using Goldmann applanation tonometer, gonioscopy, funduscopic exam and a posterior segment OCT after pupil dilatation to measure the RNFL and macular ganglion cell layer (CGL). To obtain the AS measurements, the AS module of DRI-Triton^®^ (Topcon Corporation, Tokyo, Japan) using the ‘line’ anterior segment capture mode of a 6-mm exploration field was employed. The OCT images at the 3- and 9-o’clock positions were obtained by two well-trained examiners (BK and SBR), accepting only images of good quality, as determined by the maximum of the signal strength intensity (SSI) above 40. Additionally, using the same OCT device, a 3D optic disc entailing a 6 × 6 mm raster scan centered at the optic disc and 6 × 6 mm raster scan of the macular region were used to obtain the RNFL and CGL measurements.

### 2.3. OCT Measurements

AST and conjunctival-Tenon capsule thickness (CTT) were manually measured in the b-scan OCT images. The sclera was clearly identified as hyper-reflective tissue between the internal hypo-reflective ciliary body tissue and the hypo-reflective tissue presenting the conjunctiva-Tenon capsule. The AST and CTT at 0 mm (AST0 and CTT0), 1 mm (AST1 and CTT1), at 2 mm (AST2 and CTT2) and at 3 mm (AST3 and CTT3) from the scleral spur were measured in the temporal and nasal quadrants (Figure 1).

To assess the inter-observer reliability of the AST and CTT measurements, two observers (BKB and SBR) performed the measurements in 40 randomly selected subjects. The images obtained in 40 randomly selected subjects of the sample were used. In addition, to determine intra-observer reliability, one observer (SBR) took measurements of the same 40 images two weeks after the first measurements. The same AS-OCT images were used to measure the anterior chamber angle (ACA), SC cross-sectional diameter (SCD) and TM cross-sectional diameter (TMD), as previously described in a previous study by our group using the same OCT device [20] (Figure 2).

### 2.4. Statistical Analysis

All statistical tests were performed using the software package SPSS^®^ (Statistical Package for Social Sciences, v21.0; SPSS Inc., Chicago, IL, USA). Quantitative data are provided as the mean and standard deviation. Qualitative data are expressed as their frequency distributions. The Mann–Whitney U test was employed to analyze the differences between controls and HMG and the differences between HM and HMG. Bonferroni’s method was used to correct *p* values, with *p* values < 0.025 considered significant; differences in categorical variables were compared using chi-square test considering *p* values < 0.05 significant. Correlations between AST and the remaining factors (AXL, ACD, ACA, RNFL, CGL and IOP) were assessed using Pearson’s correlation coefficients. The reliability of the AST measurements was assessed by estimating the intraclass correlation coefficients (ICC; two-way mixed effects, absolute agreement and single measurement) for intra-observer and inter-observer agreement. In this analysis, the two eyes of the same patient were included since that previous studies of AST and CTT measurements by AS-OCT showed a positive correlation between the both eyes measurements [21,22]. Furthermore, there is little penalty for use of two-eye analysis, even when the inter-eye correlation is zero. [23,24].

## 3. Results

One hundred and twenty eyes were consecutively enrolled: 40 healthy eyes of 20 control subjects, 39 eyes of 20 subjects with HM and 41 eyes of 21 subjects with HMG. All the variables analysed were normally distributed. Table 1 shows the characteristics of the population studied.

In the overall study population, the mean age was 64.2 ± 11.0 years (range 45 to 83 years). The proportion of women was 54% for the overall study sample, and 54% and 53% in the HM and HMG groups, respectively. There was no difference in either the mean age or in the sex distribution between groups (*p* > 0.025). As compared to the controls, the AXL was larger in both the HM and HMG subjects (27.6 ± 1.8 mm vs. 22.6 ± 1.2 mm and 28.5 ± 2.2 mm vs. 22.6 ± 1.2, respectively, *p* < 0.025 each, respectively), but no difference was observed between HM and HMG (27.6 ± 1.8 mm vs. 28.5 ± 2.2 mm, respectively, *p* = 0.083). Both RNFL and CGL were thinner in HMG compared to the controls and HM (*p* < 0.025 each, respectively.

As compared with both control eyes and HM eyes, the proportion of detectable SC in AS-OCT images was significantly lower in both temporal (65.6% vs. 87.5%, respectively; *p* = 0.039 and 63.3% vs. 82.6%, respectively; *p* = 0.040) and nasal quadrants (63.3% vs. 84.8%, respectively; *p* = 0.011 and 63.3% vs. 83.3%, respectively; *p* = 0.021.

In the temporal quadrant, SCD and TMD were significantly longer in the HMG subjects compared to controls (380.0 ± 62 μm vs. 316.7 ± 72 μm, *p* = 0.001) and (637.6 ± 113 μm vs. 512.1 ± 97 μm, *p* = 0.000), respectively. Nasal quadrants showed similar results. There were no significant differences in both SCD and TMD measurements between HM and HMG groups (all *p* > 0.025), (Table 2).

Compared with the control group, temporal AST0 (432.5 ± 79 μm vs. 54.7 ± 84 μm, *p* = 0.000), temporal AST1 (383.9 ± 64 μm vs. 454.4 ± 89 μm, *p* = 0.003), temporal AST2 (404.0 ± 68 μm vs. 461.1 ± 97 μm, *p* = 0.016) and temporal AST3 (403.0 ± 80 μm vs. 476.1 ± 91 μm, *p* = 0.004) were significantly thinner in the HMG group. Similarly, compared to the HM group, the temporal AST0 (432.5 ± 79 μm vs. 532.8 ± 99 μm, *p* = 0.000), temporal AST1 (383.9 ± 64 μm vs. 460.5 ± 80 μm, *p* = 0.000), temporal AST2 (404.0 ± 68 μm vs. 464.0 ± 88 μm, *p* = 0.006) and temporal AST3 (403.0 ± 80 μm vs. 458.1 ± 91 μm, *p* = 0.014) were significantly thinner in the HMG group. Nasal AST was thinner in the HMG group compared to the controls and HM groups (*p* < 0.025 each, respectively) (Table 3).

No differences were detected in the temporal and nasal CTT measurements at 0, 1, 2 and 3 mm from the scleral spur between HMG, controls and HMG groups (*p* > 0.025 each, respectively) (Table 4).

In HMG group, no significant correlation was found between AST measurements and AXL, while a mild significant correlation was found between temporal AST0, nasal AST0 and CGL measurements (R = 0.411, 0.425; *p* < 0.05 each, respectively). ACD, ACA, IOP and RNFL measurements did not correlate to AST measurements (Table 5).

In the reliability study, intra- and inter-observer agreement were excellent and very good, respectively. Inter- and intra-observer ICC were 0.915 (95% ICC 0.867–0.940) and 0.852 (95% ICC 0.821–0.910) for the AST measurements in the temporal quadrant, respectively (Table 6).

## 4. Discussion

High myopia patients were reported to be more susceptible to glaucoma than subjects without myopia [25]. Additionally, it has been reported that HM was associated with biomechanical changes in eye wall structures, including sclera [26], which is the major stress-bearing structure of the eye. In addition, these changes in the scleral biomechanical properties may firstly affect its response to the intraocular pressure at optic nerve level [27], and secondly, its role in maintaining the function of aqueous humour system, especially the Schlemm canal [10]. Moreover, unlike other eyeball wall parameters, such as the central corneal thickness, which have been previously reported to be a predictor of glaucoma development [28], little is known about the role of the AST in the pathogenesis of glaucoma. Therefore, the scleral properties, especially the anterior sclera, may be central to various aspects of glaucoma development in HM eyes.

In this study, we compared the AST between HMG subjects, HM subjects and healthy controls. It is noteworthy that the AST was thinner in all the points measured (at 1, 2 and 3 mm from the scleral spur) and in both quadrants in HMG eyes compared to both the controls and HM without glaucoma (*p* < 0.025 each, respectively), with an observed difference in AST measurements that ranged from 50 to 100 μm (10% to 20%) between HM and HMG. In agreement with the current study, Yan et al. [18] reported a thinner AST0 (728.84  ±  99.33 vs. 657.39  ±  67.02 μm, *p*  <  0.001), AST1 (537.79  ±  79.55 vs. 506.83  ±  57.37 μm, *p*  =  0.038) and AST3 (571.09  ±  79.15 vs. 532.13  ±  59.84 μm, *p*  =  0.009) in PAOG compared to controls with an AXL of 23.27  ±  0.89 mm. Additionally, in myopic eyes, Dhakal R et al. [29] found that AST was thinner compared with emmetropic eyes in temporal and nasal quadrants (583.24 ± 15.00 and 587.09 ± 27.00 μm), respectively. In Caucasians, Read et al. [30] observed thinner AST (thinner by >75 μm) in a relatively lower range of myopes (≤−8.00 D), compared with the current study (which included eyes with an AXL of 28.2 ± 2.2 mm). Similarly, Zhou et al. [31] found, in a group of younger myopic patients (32.2  ±  9.9 years), that the AST was thinner from 0 to 6 mm from scleral spur in HM eyes (AXL = 29.05  ±  2.0 mm) compared with emmetropic eyes. Absolute values of the previous studies are not comparable to the current study due mainly to the use of different AS-OCT devices with different axial penetration, which makes difficult to demarcate the boundary between the episclera and the sclera, as well as for considering the episcleral tissue as part of the measured sclera.

In relation to SC and TM, our results indicate larger SCD and TMD in both HM and HMG compared to control eyes. in concordance with our results Chen et al. [32] found that subjects with high myopia had a significantly larger SC diameter in the nasal (188.1 ± 85.6 μm vs. 127.4 ± 46.7 μm, *p* < 0.001) and temporal (200.1 ± 7.2 μm vs. 147.6 ± 68.1 μm, *p* = 0.001) quadrants, respectively. Similarly, Li et al. [11] found that SC area (6622.68 ± 1130.06 vs. 6105.85 ± 1297.84 μm^2^, *p* = 0.015) was significantly greater in the HM group than in the control group.

In the current study, no differences were observed in SCD and TMD between HM and HMG, but remarkably, SC was less detectable in HMG subjects compared to HM in both the nasal and temporal quadrants. We have speculated two possible reasons for these results. The first one is that in HMG subjects, a deformed and compressed sclera would apply more pressure to the intrascleral collector channels and deep scleral plexus, which may result in the compression and obstruction of SC. In second place, the thinner anterior scleral at the level of scleral spur may provide a lesser support to maintain open the SC. Furthermore, we think that the underlying mechanism for the difference of detectable SC between HM and HMG still needs more investigations.

Although the elongation of the eyeball in HM leads to the thinning of many ocular structures, such as posterior sclera, choroid and retina, as previously described [33,34], our results in HMG with an AXL range from 26.0 mm to 33.7 mm indicate insignificant correlation between AST measurements and AXL, ACD and ACA (*p* > 0.05). In accordance, Li et al. [11], in their study of high myopic eyes (26.68 ± 0.96), found no significant correlation (*p* > 0.05) between AXL and AST0,1,2,3. Similarly, Buckhurst et al. [35] found that AXL, in cases of AXL > 25.5, AST3 mm failed to show any significant effect on AST (*p* = 0.907).

Both results of the current study and those of the previous ones may indicate that there might be other factors affecting the anterior scleral thinning rather than ocular expansion during the axial elongation, when AXL is more than 26 mm. Additionally, we found no significant difference in other baseline parameters between HM and HMG eyes, with the exception of RNFL thickness, this observation suggests that AST may be an independent factor, distinct from other factors related to AXL, such as ACD and ACA. This could imply that AST is more closely related to glaucoma than other factors.

Surprisingly, we found a mild correlation between temporal AST0, nasal AST0 and CGL measurements (R = 0.411, 0.425, *p* < 0.05 each, respectively), while no correlation was found with RNFL thickness (*p* > 0.05). Conversely, Seol et al. [36] reported that CGL measurement was the best parameter for discriminating between myopic eyes with glaucoma and myopic eyes without glaucoma with AUROCs 0.752 (95% CI 0.692–0.811) compared to AUROCs of average RNFL thickness 0.677 (95% CI 0.614–0.740). In our study, this correlation with CGL may suggest that the AST0, which reflects the scleral thickness most proximate to SC and TM and is probably responsible for maintaining their function, might be a novel biomarker to discriminate glaucoma from healthy in high myopic subjects.

Finally, no differences were observed in CTT between the three groups. In agreement with our results, Fernandez-Vigo et al. [21], in a study that included 630 participants, observed no influence by refractive error and AST on the CTT measurements (*p* > 0.065). Additionally, Read et al. [30] found that anterior conjunctival thickness was not significantly associated with AXL in healthy subjects. According to these findings we think that the conjunctiva plays a negligible or non-existent role in the pathogenesis of glaucoma in patients with high myopia.

This study has several limitations. The first limitation is that, to determine AST measurements, only horizontal quadrants were assessed but not the vertical ones. This means that a pressure on the lid necessary to visualize the vertical sclera might modify the measurements. Similarly, the identification of Schlemm’s canal proved to be challenging in some cases, leading to an inability to detect it in all cases. This, in turn, may impact our results. Additionally, all the subjects included in the current study were Caucasians. Therefore, caution should be exercised when generalizing these findings to other racial groups. Another limitation in our study is that a majority of HMG participants were treated with prostaglandin analogues (PGAs). Notably, prior studies have suggested a potential impact of PGAs on anterior segment structures in glaucoma patients [37,38]. However, future longitudinal studies with larger samples are needed to prove these effects and to determine the requisite time for manifestation in myopic patients.

In this study, the pachymetry was not recorded. We believe corneal thickness could be an indicator of change in anterior segment biometrics in HMG subjects and correlated to AST. Finally, this is a cross-sectional study with a relatively limited sample. Further prospective and longitudinal studies are needed to confirm these results.

## 5. Conclusions

In summary, anterior segment SS-OCT enables the detection of changes in scleral thickness and visibility of Schlemm’s canal in eyes with high myopia and in eyes with high myopia and glaucoma. These findings underscore the importance of future research to delve deeper into whether such changes are an outcome of the glaucoma disease or may serve as risk indicators for the development of glaucoma in this population.

## Figures and Tables

**Figure 1 diagnostics-14-00655-f001:**
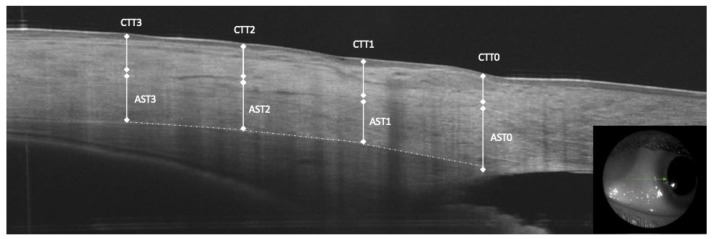
Anterior segment optical coherence tomography measurements of anterior scleral thickness (AST) at 0, 1, 2 and 3 mm from the scleral spur, conjunctiva and tenon capsule thickness (CTT).

**Figure 2 diagnostics-14-00655-f002:**
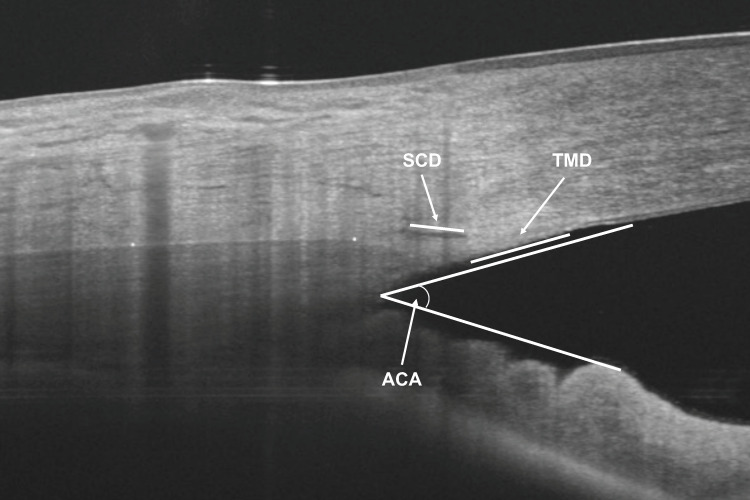
Anterior segment optical coherence tomography measurements of anterior chamber angle (ACA), Schlemm’s canal cross sectional diameter (SCD) and trabecular meshwork cross sectional diameter (TMD).

**Table 1 diagnostics-14-00655-t001:** Characteristics of the population studies. Mean ± standard deviation (range).

Variable	Controls	HM	HMG	P1	P2
Age (years)	63.5 ± 12.7 (46–83)	63.7 ± 10.8 (41–80)	62.6 ± 10.3 (43–81)	0.635	0.733
AXL (mm)	22.6 ± 1.2 (21.3–23.9)	27.6 ± 1.8 (26.0–32.9)	28.2 ± 2.2 (26.0–33.7)	0.000	0.083
ACD (mm)	3.1 ± 0.3	3.4 ± 0.5	3.55 ± 0.8	0.000	0.026
Temporal ACA (degree)	27.8 ± 6 (20–42)	36.9 ± 9 (33–48)	35.6 ± 5 (31–48)	0.000	0.437
Nasal ACA (degree)	30.2 ± 8 (21–49)	39.2 ± 12 (32–47)	39.7 ± 5 (31–51)	0.000	0.706
RNFL (µm)	101.0 ± 10 (95–128)	85.4 ± 19 (82–106)	69.0 ± 28 (66–82)	0.000	0.124
CGL+ (µm)	67.2 ± 16 (55–77)	60.6 ± 12 (31–70)	55.4 ± 13 (33–65)	0.024	0.022
IOP (mmHg)	14.2 ± 1 (10–17)	13.3 ± 3 (10–16)	13.6 ± 1 (10–15)	0.341	0.338

HM: high myopia; HMG: high myopic glaucoma; ACD: anterior chamber depth; AXL: axial length; ACA: anterior chamber angle; RNFL: retinal nerve fibre layer; CGL: ganglion cell layer; IOP: intraocular pressure; P1: The Mann–Whitney U test comparing HMG and controls; P2: The Mann–Whitney U test comparing HMG and HM.

**Table 2 diagnostics-14-00655-t002:** Characteristics of Schlemm’s canal and trabecular meshwork. Mean ± standard deviation (range).

Variable	Controls 40	HM 39	HMG 41	P1	P2
Temporal SCV	35 (87.5%)	32 (82.0%)	26 (63.4%)	0.039 ^b^	0.040 ^b^
Temporal SCD (µm)	316.7 ± 72 (174–492)	361.3 ± 59 (249–507)	380.0 ± 62 (276–574)	0.001	0.306
Temporal TMD (µm)	512.1 ± 97 (387–673)	634.3 ± 139 (348–715)	637.6 ± 113 (443–857)	0.000	0.652
Nasal SCV	34 (85.0%)	32 (82.0%)	25 (60.9%)	0.011 ^b^	0.021 ^b^
Nasal SCD (µm)	342.9 ± 88 (220–493)	370.8 ± 65 (252–496)	378.2 ± 68 (227–494)	0.022	0.268
Nasal TMD (µm)	523.6 ± 106 (377–660)	614.5 ± 123 (433–720)	627.4 ± 90 (433–786)	0.001	0.435

HM: high myopia; HMG: high myopic glaucoma; SCV: number of cases with detectable of Schlemm canal; SCD: Schlemm’s canal cross-sectional diameter; TMD: trabecular meshwork cross-sectional diameter; P1: The Mann–Whitney U test comparing HMG and controls; P2: The Mann–Whitney U test comparing HMG and HM. ^b^: chi-square test.

**Table 3 diagnostics-14-00655-t003:** Anterior scleral thickness of different measurement locations from the scleral spur. Mean ± standard deviation (range).

Variable	Controls	HM	HMG	P1	P2
Temporal AST0 (µm)	547.7 ± 84 (433–727)	532.8 ± 99 (338–847)	432.5 ± 79 (312–619)	0.000	0.000
Temporal AST1 (µm)	454.4 ± 89 (325–638)	460.5 ± 80 (360–709)	383.9 ± 64 (259–543)	0.003	0.000
Temporal AST2 (µm)	461.1 ± 97 (261–675)	464.0 ± 88 (328–736)	404.0 ± 68 (282–544)	0.016	0.006
Temporal AST3 (µm)	476.1 ± 91 (319–695)	458.1 ± 91 (276–748)	403.0 ± 80 (258–589)	0.004	0.014
Nasal AST0 (µm)	527.2 ±86 (412–799)	524.7 ± 101 (334–779)	449.8 ± 91 (309–701)	0.001	0.002
Nasal AST1 (µm)	479.0 ± 81 (343–675)	498.8 ± 101 (380–840)	432.8 ± 87 (270–644)	0.032	0.016
Nasal AST2 (µm)	504.4 ± 87 (337–748)	509.8 ± 97 (388–832)	448.7 ± 95 (286–648)	0.012	0.016
Nasal AST3 (µm)	501.1 ± 128 (250–765)	496.7 ± 104 (356–871)	440.6 ± 89 (287–656)	0–035	0.054

HM: high myopia; HMG: high myopic glaucoma; AST: Anterior scleral thickness; P1: The Mann–Whitney U test comparing HMG and controls; P2: The Mann–Whitney U test comparing HMG and HM.

**Table 4 diagnostics-14-00655-t004:** Conjunctiva and tenon capsule thickness of different measurement locations from the scleral spur. Mean ± standard deviation (range).

Variable	Controls	HM	HMG	P1	P2
Nasal CTT0 (µm)	251.9 ± 70 (164–535)	247.0 ± 59 (164–432)	249.2 ± 55 (175–447)	0.962	0.833
Nasal CTT1 (µm)	244.0 ± 95 (118–646)	227.2 ± 60 (142–395)	241.2 ± 52 (162–365)	0.606	0.202
Nasal CTT2 (µm)	238.5 ± 75 (146–529)	217.5 ± 58 (113–332)	235.5 ± 46 (163–321)	0.757	0.091
Nasal CTT3 (µm)	220.8 ± 63 (132–450)	214.7 ± 50 (100–288)	225.2 ± 47 (157–339)	0.411	0.670
Temporal CTT0 (µm)	245.4 ± 46 (141–338)	233.7 ± 55 (124–405)	246.6 ± 53 (164–410)	0.851	0.365
Temporal CTT1 (µm)	226.5 ± 46 (130–316)	226.6 ± 49 (152–364)	221.3 ± 31 (165–307)	0.653	0.904
Temporal CTT2 (µm)	216.0 ± 39 (124–308)	215.1 ± 59 (138–418)	216.3 ± 45 (148–327)	0.682	0.600
Temporal CTT3 (µm)	206.1 ± 37 (130–271)	201.7 ± 53 (111–399)	213.9 ± 43 (135–307)	0.669	0.251

HM: high myopia; HMG: high myopic glaucoma; CTT; conjunctiva tenon capsule thickness; P1: The Mann–Whitney U test comparing HMG and controls; P2: The Mann–Whitney U test comparing HMG and HM.

**Table 5 diagnostics-14-00655-t005:** Correlation between anterior scleral thickness and the other anterior segment biometrics.

Variable	AXL (mm)	ACD (mm)	Temporal ACA (Degree)	Nasal ACA (Degree)	RNFL (µm)	CGL (µm)	IOP (mmHg)
Nasal AST0 (µm)	−0.192 (*p* = 0.336)	0.311 (*p* = 0.139)	0.244 (*p* = 0.251)	0.014 (*p* = 0.947)	−0.114 (*p* = 0.725)	0.425 (*p* = 0.043)	−0.234 (*p* = 0.525)
Nasal AST1 (µm)	−0.143 (*p* = 0.476)	0.277 (*p* = 0.190)	0.202 (*p* = 0.343)	0.044 (*p* = 0.837)	−0.320 (*p* = 0.311)	0.220 (*p* = 0.516)	−0.323 (*p* = 0.331)
Nasal AST2(µm)	−0.186 (*p* = 0.352)	0.253 (*p* = 0.233)	0.203 (*p* = 0.340)	−0.037 (*p* = 0.862)	−0.306 (*p* = 0.333)	0.214 (*p* = 0.528)	−0.206 (*p* = 0.363)
Nasal AST3 (µm)	−0.279 (*p* = 0.158)	0.128 (*p* = 0.551)	0.111 (*p* = 0.606)	−0.009 (*p* = 0.966)	−0.275 (*p* = 0.387)	−0.261 (*p* = 0.438)	−0.175 (*p* = 0.387)
Temporal AST0 (µm)	−0.192 (*p* = 0.168)	0.311 (*p* = 0.069)	0.244 (*p* = 0.125)	0.014 (*p* = 0.474)	−0.114 (*p* = 0.363)	0.411 (*p* = 0.046)	−0.134 (*p* = 0.363)
Temporal AST1 (µm)	−0.143 (*p* = 0.238)	0.277 (*p* = 0.095)	0.202 (*p* = 0.171)	0.044 (*p* = 0.419)	−0.320 (*p* = 0.155)	0.220 (*p* = 0.258)	−0.220 (*p* = 0.155)
Temporal AST2 (µm)	−0.186 (*p* = 0.176)	0.253 (*p* = 0.116)	0.203 (*p* = 0.170)	−0.037 (*p* = 0.431)	−0.306 (*p* = 0.167)	0.214 (*p* = 0.264)	−0.206 (*p* = 0.147)
Temporal AST3 (µm)	−0.279 (*p* = 0.079)	0.128 (*p* = 0.276)	0.111 (*p* = 0.303)	−0.009 (*p* = 0.483)	−0.275 (*p* = 0.194)	−0.261 (*p* = 0.219)	

AST: Anterior scleral thickness; ACD: anterior chamber depth; AXL: axial length; ACA: anterior chamber angle; RNFL: retinal nerve fibre layer; CGL: ganglion cell layer; IOP: intraocular pressure.

**Table 6 diagnostics-14-00655-t006:** Reliability of anterior scleral measurements by optical coherence tomography in high myopic glaucoma group. Mean ± standard deviation (range).

HMG	Temporal AST	Nasal AST
Observer 1: First measurement	405.0 ± 72 (259–619)	442.8 ± 90 (286–701)
Observer 1: Second measurement	421.0 ± 65 (267–607)	455.8 ± 93 (280–692)
Observer 2	441.0 ± 63 (261–599)	433.8 ± 84 (279–682)
Intraobserver ICC	0.915 (95% ICC 0.867–0.940)	0.918 (95% ICC 0.867–0.940)
Interobserver ICC	0.852 (95% ICC 0.821–0.910)	0.912 (95% ICC 0.821–0.930)

HMG: High myopic glaucoma; ICC: intraclass correlation coefficient (95% confidence interval); AST: anterior scleral thickness.

## Data Availability

Data used to support the findings presented in this study are available on request from the corresponding author.

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
