# Peer review of "Anterior Scleral Thickness and Anterior Segment Biometrics Measured with Swept Source Ocular Coherence Tomography in High Myopic Eyes with and without Glaucoma: A Comparative Study"

_diagnostics, 2024, doi:10.3390/diagnostics14060655_

Round 1

Reviewer 1 Report

Comments and Suggestions for Authors

The authors present a study designed evaluate and compare the biometrics of the anterior segment between high myopia subjects, with (HMG) and without glaucoma (HM), and control eyes without myopia, with a focus in the anterior sclera thickness. The study may deserve publication, but several issues must be addressed before the manuscript can be accepted.

Something must have occurred when producing figures 1 and 2 because the OCT B-Scans are not visible at all. The original images document, uploaded with the article, only contains the B-Scan of Figure 1.

Moreover, in Figure 2 it is possible to read the subject name, which is not acceptable.

In tables 1-4 the authors present comparisons between the HM and the control groups and between the HMG and the control groups. In the text, are also presented the results from comparisons between the HM and HMG groups. This indicates that multiple comparisons between groups were performed. The authors also state that used the paired t test for comparisons between groups. Here I have three concerns:

1 - I do not understand why the paired-sample t test was used. This test concerns situations where each observation in one sample is, in some way, correlated with an observation in the second sample. However, in my understanding, the 3 samples (HMG, HM and Controls) are independent samples.

2 - Performing multiple comparisons between with the t-test is invalid and increases greatly the probability of committing a type I error (rejecting the null hypothesis when it is true). Multiple comparisons with the t-test require a correction on the p value for maintaining the 5% level of significance, like the Bonferroni correction. A more robust and recommended methodology is to perform an ANOVA analysis with a post-hoc comparison test, like the Tukey test.

3 - The t-test (and the ANOVA) requires equal variances (homogeneity of variance) in the samples being compared. In some measurements, there are clear differences in the standard deviation of the 3 samples (for example, Nasal ACA, RNFL, Temporal TMD and others). Although the t-test (and the ANOVA) is robust enough to considerable departures from its assumptions, particularly when the samples’ size is large and equal, the inhomogeneity of variance increases the probability of a type I error. This issue should be verified (using, for example, the Welch’s approximate t) or at least acknowledged in the Discussion.

Although the interclass coefficient correlation allows to assess the reliability of the measurements, it would be much more valuable to provide the maximum difference between repeated measurements for a given level of confidence. For a confidence of 95%, this corresponds to 2.77 times the intra-subject standard deviation (SW). The square of the SW is the within-groups mean square, a parameter that is usually provided by ANOVA analysis. This is just a suggestion to the authors.

In Table 2, I don’t understand why the chi-square test was used for the Temporal and Nasal SCV parameters. As far as I can see, they are not categorical variables.

The caption of Table 6 uses the term “Reproducibility” for the reliability assessment. Currently, in Metrology, it is internationally accepted that reproducibility implies different measurement systems between trials, which is not the case here. In my opinion, it would be better to use just the term “reliability”, which is not an “official” term in Metrology. Using “repeatability” is not an alternative option since it implies measurements by the same observer and, therefore, does not cover the inter-observer variability.

Minor issues:

There is something wrong in line 118/119: “performed the measurements in 40 randomly selected subjects the images obtained in 30 randomly selected subjects of the sample were used”.

There is also a mistake in line 145: “This. One hundred and twenty…”.

Author Response

Response to reviewer: Thank you for your thorough review and valuable observations. We truly appreciate your diligence in highlighting the significance of the statistical analysis in our study. Your feedback has been helpful in improving the quality of our work.

In response to your concerns, the statistical analysis has been meticulously revised and redone under the supervision of a professional with expertise in the field. This individual, who oversees studies at our university hospital, has ensured that all modifications align with your comments and observations. The revisions have been appropriately incorporated into the methods, results, and discussion sections of our manuscript.

We acknowledge the importance of your input, and we are committed to achieving the highest standards in our research. If, upon reevaluation, you believe that further adjustments are necessary, we are more than willing to collaborate with our statistical expert to implement any additional changes as per your recommendations.

The authors present a study designed evaluate and compare the biometrics of the anterior segment between high myopia subjects, with (HMG) and without glaucoma (HM), and control eyes without myopia, with a focus in the anterior sclera thickness. The study may deserve publication, but several issues must be addressed before the manuscript can be accepted.

Something must have occurred when producing figures 1 and 2 because the OCT B-Scans are not visible at all. The original images document, uploaded with the article, only contains the B-Scan of Figure 1.

Moreover, in Figure 2 it is possible to read the subject name, which is not acceptable.

Response to reviewer: Both figures have been adjusted and reviewed to eliminate technical errors.

In tables 1-4 the authors present comparisons between the HM and the control groups and between the HMG and the control groups. In the text, are also presented the results from comparisons between the HM and HMG groups. This indicates that multiple comparisons between groups were performed. The authors also state that used the paired t test for comparisons between groups. Here I have three concerns:

1 - I do not understand why the paired-sample t test was used. This test concerns situations where each observation in one sample is, in some way, correlated with an observation in the second sample. However, in my understanding, the 3 samples (HMG, HM and Controls) are independent samples.

2 - Performing multiple comparisons between with the t-test is invalid and increases greatly the probability of committing a type I error (rejecting the null hypothesis when it is true). Multiple comparisons with the t-test require a correction on the p value for maintaining the 5% level of significance, like the Bonferroni correction. A more robust and recommended methodology is to perform an ANOVA analysis with a post-hoc comparison test, like the Tukey test.

3 - The t-test (and the ANOVA) requires equal variances (homogeneity of variance) in the samples being compared. In some measurements, there are clear differences in the standard deviation of the 3 samples (for example, Nasal ACA, RNFL, Temporal TMD and others). Although the t-test (and the ANOVA) is robust enough to considerable departures from its assumptions, particularly when the samples’ size is large and equal, the inhomogeneity of variance increases the probability of a type I error. This issue should be verified (using, for example, the Welch’s approximate t) or at least acknowledged in the Discussion.

Response to reviewer: 

Our study incorporated three distinct samples of relatively small sizes. Consequently, to analyse the differences between the control group and HMG, as well as between HMG and HM, we utilized the Mann-Whitney U test. This non-parametric test was chosen because it doesn't require the assumption of variance homogeneity. Due to the extensive range of variables, we applied Bonferroni's method for adjusting p-values, setting the significance threshold at p < 0.025.

Although the interclass coefficient correlation allows to assess the reliability of the measurements, it would be much more valuable to provide the maximum difference between repeated measurements for a given level of confidence. For a confidence of 95%, this corresponds to 2.77 times the intra-subject standard deviation (SW). The square of the SW is the within-groups mean square, a parameter that is usually provided by ANOVA analysis. This is just a suggestion to the authors.

Response to reviewer:  we wish to retain the analysis involving the interclass correlation coefficient. The primary objective of our study is to identify differences in anterior segment biometrics among HMG, HM, and control groups. This aims to uncover potential associations with the glaucoma mechanism within the high myopic population. The reliability of the Anterior Segment Tomography (AST) measurements was evaluated by calculating intraclass correlation coefficients, both for intra-observer and inter-observer agreement, ensuring the precision of our findings. Additionally, the reliability of these measurements has been previously established through our participation in larger sample studies. This further substantiates the validity of our methodological approach and the significance of our findings.

In Table 2, I don’t understand why the chi-square test was used for the Temporal and Nasal SCV parameters. As far as I can see, they are not categorical variables.

Response to reviewer:  In this comparison we compared the percentage of cases were the Schlemm canal was detected, considering this variable as a categorical one rather than numerical since it has two options visible or not visible, for this reason we applied chi-square test.

The caption of Table 6 uses the term “Reproducibility” for the reliability assessment. Currently, in Metrology, it is internationally accepted that reproducibility implies different measurement systems between trials, which is not the case here. In my opinion, it would be better to use just the term “reliability”, which is not an “official” term in Metrology. Using “repeatability” is not an alternative option since it implies measurements by the same observer and, therefore, does not cover the inter-observer variability.

Response to reviewer:  as recommended by the reviewer we change reproducibility to reliability in the table 6 caption and in the main text.

Change in the manuscript:

In the s Reliability study, intra and inter-observer agreement were excellent and very good, respectively. Inter- and intra-observer ICC were 0.915 (95% ICC 0.867-0.940) and 0.852 (95% ICC 0.821-0.910) for the AST measurements in the temporal quadrant, respectively. (Table 6).

Table 6. Reliability of anterior scleral measurements by optical coherence tomography in high myopic glaucoma group. Mean ± standard deviation (range).

Minor issues:

There is something wrong in line 118/119: “performed the measurements in 40 randomly selected subjects the images obtained in 30 randomly selected subjects of the sample were used”. 

There is also a mistake in line 145: “This. One hundred and twenty…”.

Response to reviewer:  Both mistakes were corrected in the manuscript.

Reviewer 2 Report

Comments and Suggestions for Authors

1. The important information about intra-ocualr pressure and pacymetry is lacking. How were these parameters related to the outcomes variables, should be briefed.

2. Except for RNFL thickness, there was no significant difference in other baseline parameters (like axil length, GCC thickness etc) between HM and HMG eyes. Contaraily, the outcome variables are significantly different? Authors should provide probable explananation for this.   

3. The conclusions drwan "....SS OCT can provide a non-invasive evaluation of the aqueous humour drainage system in high myopic eyes," is not appropriate.  The study indicates changes observed in high myopic eyes and high myopic eyes with glaucoma. But does the result provide any significant predictability for future development of glaucoma, is not clear. The observed changes could be outcomes of diseases rather than precedding changes. The correction in conclusions is required. 

Author Response

Response to reviewer: Thank you for your thorough review and valuable observations. We truly appreciate your work.  Your feedback has been helpful in improving the quality of our work.

  1. The important information about intra-ocualr pressure and pacymetry is lacking. How were these parameters related to the outcomes variables, should be briefed.

Response to the reviewer: In our study, we did not measure pachymetry due to the fact that some patients across all three groups had previously undergone laser refractive surgery. Importantly, this was not considered an exclusion criterion, as no existing studies have established a definitive link between laser refractive surgery and alterations in scleral measurements. However, it's worth noting that these past surgeries, particularly those with a high ablation profile, may have led to reduced pachymetry measurements and potentially impacted intraocular pressure (IOP) readings. Consequently, the comparison between groups and the examination of the correlation between pachymetry and anterior segment thickness (AST) measurements could be influenced by the modified pachymetry values.

Moreover, in the case of the HMG, the IOP values are influenced by ongoing glaucoma treatment, introducing a potential confounding factor in the relationship between outcome variables. Despite these considerations, we retrospectively collected IOP measurements for all three groups, incorporated the values into Table 1, and computed the correlation between AST and IOP, which we included in Table 5 and the Results section. Additionally, we acknowledged the absence of actual pachymetry measurements as a limitation in our study.

Changes in manuscript: discussion

Another limitation in our study is that a majority of HMG participants were treated with prostaglandin analogues (PGAs). Notably, prior studies have suggested a potential impact of PGAs on anterior segment structures in glaucoma patients [37,38]. However, future longitudinal studies with larger samples are needed to prove these effects and to determine the requisite time for manifestation in myopic patients.

In this study the pachymetry was not recorded, we believe corneal thickness could be an indicator of change in anterior segment biometrics in HMG subjects and correlated to AST. Finally, this is a cross-sectional study with a relatively limited sample. Further prospective and longitudinal studies are needed to confirm these results

  1. Except for RNFL thickness, there was no significant difference in other baseline parameters (like axil length, GCC thickness etc) between HM and HMG eyes. Contaraily, the outcome variables are significantly different? Authors should provide probable explananation for this. 

Response to the reviewer: We probably suggest that anterior AST is more related to glaucoma than to myopia like other parameters like ACD or ACA.

Changes in manuscript: discussion

Both, the results of the current study and those of the previous ones, may indicate that there might be other factors affecting the anterior scleral thinning rather than ocular expansion during the axial elongation when AXL is more than 26 mm.

Additionally, we found no significant difference in other baseline parameters between HM and HMG eyes, with the exception of RNFL thickness, this observation suggests that AST may be an independent factor, distinct from other factors related to AXL, such as ACD and ACA. This could imply that AST is more closely related to glaucoma than other factors.

  1. The conclusions drwan "....SS OCT can provide a non-invasive evaluation of the aqueous humour drainage system in high myopic eyes," is not appropriate.  The study indicates changes observed in high myopic eyes and high myopic eyes with glaucoma. But does the result provide any significant predictability for future development of glaucoma, is not clear. The observed changes could be outcomes of diseases rather than precedding changes. The correction in conclusions is required. 

Response to the reviewer: As recommended by the reviewer the occlusion section was corrected.

Changes in manuscript: conclusions

In summary, anterior segment SS OCT enables the detection of changes in scleral thickness and visibility of Schlemm's canal in eyes with high myopia and in eyes with high myopia and glaucoma. These findings underscore the importance of future research to delve deeper into whether such changes are an outcome of the glaucoma disease or may serve as risk indicators for the development of glaucoma in this population.

Round 2

Reviewer 1 Report

Comments and Suggestions for Authors

The authors did a good jobr improving their manuscript and addressed properly all my questions and remarks. In my opinion, the paper can be accepted for publication.